# Understand Delegates Risk Attitudes and Behaviour: The Moderating Effect of Trust in COVID-19 Vaccination

**DOI:** 10.3390/ijerph20053936

**Published:** 2023-02-22

**Authors:** Songhong Chen, Jian Ming Luo

**Affiliations:** 1Faculty of Hospitality and Tourism Management, Macau University of Science and Technology, Macau, China; 2Faculty of International Tourism and Management, City University of Macau, Macau, China

**Keywords:** Macau, COVID-19, trust in vaccination, risk attitude, involvement, satisfaction, loyalty

## Abstract

The continuing COVID-19 pandemic has prompted many people to receive the needed vaccines. However, how trust in vaccination affects the attitude and behavior of delegates attending a convention in Macau has yet to be determined. Accordingly, we applied quantitative method in conducting a survey involving 514 participants and analysed the data using AMOS and SPSS. The results showed that trust in vaccines has a significant impact on moderating the relationship between risk attitude and satisfaction. Trust in vaccines has a significant positive effect on involvement. Risk attitude negatively affects involvement, satisfaction, and loyalty. The major contribution of this research is the introduction of a model based on trust in vaccination. To boost delegates’ confidence to attend convention activities, governments and organizations should deliver accurate information on vaccines and pandemic risks, and that delegates should obtain accurate information about it. Lastly, unbiased and professional operators of the MICE industry also can offer precise COVID-19 vaccination information to reduce misperception and increase the security.

## 1. Introduction

Trade and associated economic activities have been negatively affected by the global healthcare and economic crisis [1]. Globally and domestically in China, the meetings, incentives, conventions, and exhibitions (MICE) industry is growing faster in the tourism area [2,3,4]. Since the severe acute respiratory syndrome coronavirus (COVID-19) outbreaks globally, quarantine regulations and border closures hit the worldwide MICE industry [5]. According to the ICCA report, there were only 4843 events of MICE in 2020, the drop rate of which was 36.55% compared to 2019. In one of the largest MICE-hosting regions, Asia, the number of events has decreased by 17.85% compared to 2019. In Macau, over 500 exhibitions and conventions have been either cancelled or postponed [6]. The total revenue of the MICE sector in Macau has decreased to MOP 14.2 million from MOP 104.7 million in 2019 [7]. In the second quarter of 2022, only 91 MICE events were held in Macau, a decrease of 56 year-on-year; excluding financial support from government or organisations, these exhibitions recorded a loss of MOP 0.81 million [8].

According to the information from the Macau government, people wishing to attend selected Macau events should have completed their two-dose vaccination and present valid certification of nucleic acid test [9]. Currently, Macau has 12 medical facilities with the capacity to offer vaccination to 5000 people on a daily basis [10]. More importantly, the safety of COVID-19 vaccination, or the trust in COVID-19 vaccination, affects delegates’ physical and psychological hopes, inspiring them and ensuring security in open areas, especially when they take part in events or fairs. The annual Macau International Trade and Investment Fair (MIF) is one of the most representative international economic and trade events [11]. However, incorrect information about COVID-19 still misleads people, making the situation worse [11]. People might consider that taking part in any public activity will be more risky and dangerous, and what kind of issues will affect peoples’ attitude and behavior is becoming an interesting academic topic. Previous research has shown that trust in COVID-19 vaccines can affect people’s behavior [12]. Gursoy utilized monthly data collected from US consumers and examined the impact of consumers’ attitudes towards COVID-19 vaccination on their intentions to travel to a destination or stay in hotels [13]. The study found that people’s health related to COVID-19 has a positive impact on their ascribed responsibility to adapt emotionally (pride, guilt). Moreover, such pro-social behaviors acting on personal norms, impacted vaccination-friendly and altruistic behavior [14].

The theory of planned behavior (TPB) is the most comprehensive theory in social psychology in determining individual behavioural intention. The theory of planned behavior (TPB) is suitable to predict and explain potential behavior associated with energy use by identifying factors that impact behavior, the purpose of which is to develop behavior change programs in the tropical climate context [15,16,17,18]. Nevertheless, knowing the definition of trust remains a dynamic process [19,20,21,22,23,24,25]. The satisfaction–loyalty theory was constructed to investigate sources of customers’ loyalty to specific services, and has been extensively adapted and applied in a large area [26]. Shinyong Jung confirmed that there was a strong relationship between attendee satisfaction and loyalty in the convention sector [27]. Jinge Yao found that the attendees’ attitudes were positively impacted by the comparative advantage, perceived safety threat, outcome expectations and trialability regarding online MICE during the COVID-19 pandemic [28]. However, what constitutes value for convention attendees and how it influences outcomes is still undiscovered [29]. Furthermore, lacking a theory of planned behavior and trust theory to analyse the intention to attend online MICE events, the accurate identification of unique factors that impact intentions in the MICE sector under COVID-19 are still uninvestigated [28]. Placing greater importance on the theory of planned behavior (TPB) offers the possibility to make further distinctions among related dispositions and other kinds of beliefs. It is open to the inclusion of additional forecasters if a significant proportion of the captured variance in intention or behavior can be shown after the theory’s current variables have been explained [16]. Trust often manifests in dyadic relationships rather than across networks [29]. In festival tourism, the festival’s activities, sales concessions, environment, socialization throughout the event, authenticity, and uniqueness contribute to creating satisfaction and loyalty [27]. In spite of the speedy development and investment in the MICE industry, identifying the unique factors that create satisfaction and loyalty in the MICE sector has become meaningful under COVID-19. From a behavioural perspective, less attention has been paid to applying TPB to evaluate individual behavior based on the individual’s trust in COVID-19 vaccination. Most importantly, how the unique factor of trust under COVID-19 affects the individuals’ behavior, attitudes, satisfaction and decision to travel is still less discussed.

With the intention of discourse on these subjects, this article constructs a theoretical model to investigate the how attendees’ trust in vaccination affects attendees’ attitudes and behaviors when taking part in MICE events under the pandemic of COVID-19. The theory of planned behavior (TPB), standard satisfaction–loyalty theory, and trust theory are presented as the conceptual framework in the study. The standard satisfaction–loyalty theory identifies the relationship between delegates’ satisfaction and loyalty, trust theory identifies the delegate’s trust in vaccination, while TPB identifies the how the delegate’s trust in vaccination and the delegate’s risk attitude affect the delegate’s involvement in MICE events.

The structure of this study is as follows: Section 2 contains a literature review describing the fundamental theories that are applied, and the hypotheses that are made. Section 3 (Methodology) explains the methodology and data collection. Section 4 (Data Analysis) defines the testing hypothesis, and Section 5 (Discussion) describes the outcomes. Lastly, Section 6 (Conclusions) reviews the contributions and limits of this study.

## 2. Literature Review

### 2.1. Theoretical Background

The theory of planned behavior (TPB) is an addition which is based on the theory of reasoned action (TRA) [30]. The TPB is a framework to understand and predict consumers’ behavior. More performance will be created by a stronger intention to engage in a behavior. The assumption is that purposes determine the motivational factors that impact behavior [16]. The TPB is the most comprehensive theory in social psychology for determining behavioural intention. In this model of TPB, three basic determinants function to decide a person’s intention: perceived behavior control (PBC), subjective norms (SN), and attitudes (ATD) [15]. The TPB offers a robust theoretical basis for examining variables other than classical variables. For example, the variable “trust” is inherent in online transactions and has been studied frequently based on the TPB model [31,32,33,34,35]. 

Henning-Thurau and Klee studied solving the relationship between customer satisfaction and customer loyalty. Customer loyalty of is achieved through sustained customer satisfaction, and satisfaction of customer is complemented by an emotional component to the relationship formed with the service provider that creates consistency and willingness in patronage, premium and preference [36].

Trust is a complicated factor that concerns the relationship between individuals and organizations [37]. The different definitions of trust, including a psychological state, attitude, belief, confidence, expectation, feeling, willingness, or intention, have led to numerous misunderstandings, communication breakdowns, and confusion among researchers [38,39]. 

Based on the three theories above, the study offers a modified theoretical model which combines TPB, satisfaction–loyalty theory, and trust theory to investigate antecedents affecting delegates’ behavior in the convention sector.

### 2.2. Hypothesis Construction

#### 2.2.1. Trust 

Cognitive-based trust increases when repeated interactions allow parties to understand, predict and know processes and routines of interaction [40]. Cognitive trust occurs when a conscious decision is made based on the best knowledge [41]. When the relationships are based on cognitive trust, people choose to accept evidence of trustworthiness. 

Studies have considered that cognitive trust is often developed based on individuals’ proven reliability [41,42]. An example of a basis for building cognitive trust is a project manager’s ability in a client–project manager relationship. Confidence in vaccines relies on trust in many issues, such as science, the healthcare system, and the socio-political context. Trust is considered as an exchange for a reduction in decision complexity, a relationship between people and a system [43]. Given that trust also enhances science communication, enhancing general awareness of vaccines may be appropriate to control COVID-19 [40].

Degree of trust in vaccination is composed of two parts, namely, government and healthcare officials, both of which have been reported by interviewees using a five-point Likert scale (1 = Not at all, 5 = Completely) [44]. Trust is a possible factor that health officials identify as possibly reducing peoples’ perception of vaccination risks [45]. Another research indicated that due to a heterogeneous range of variables united with vaccine intention hesitancy, additional methods to resolve fears of vaccination are called for aside from campaigns of biased information. Government and health officials presenting promising approaches for controlling vaccination hesitation is a good response to affective reactions to community heads [46].

#### 2.2.2. Risk Attitude

Human behavior is a stable psychological dimension that can be affected and predicted by attitude [47]. Risk attitude is a mindset on risk-taking behavior [48]. Under risky circumstances, people will make decisions rationally and consider risks carefully [49]. Completely avoiding risks is possible but only occasionally [49]. According to the nature of risks, people often formulate flexible or back-up plans [50]. Risk is a subjective assessment of the probability of incurring a loss [51,52].

Risk attitude was defined as consumers’ willingness to accept risks or different risk levels, and is a consistent tendency when consumers are faced with a choice [53]. Risk attitude is an intrinsic risk-selection attribute [54]. 

#### 2.2.3. Involvement

An individuals’ involvement means the motivation and spirit to promote a specific object to a group of interest, such as an event [55]. Individuals’ involvement often relates to tourism sector research, among various involvement concepts [55,56]. Involvement is also defined as individual interests, needs, and values that urge people to participate in the pursuit of an object [57]. People who participate in the pursuit of special objects, such as products or services, will create an enhanced perception related to this object, thereby leading people to focus more on it. Therefore, people would often like to devote considerable time to acquire information about the product or service, thereby attracting people to buy it [58]. A consumer’s involvement relates to personal relevance; high-involvement purchases are very important for consumer products [59]. Although there are different definitions of involvement, some researchers have argued the importance of consumers relating with products or service offerings [59,60].

People who have received COVID-19 vaccines believe that they are effective and safe, particularly given that most of them even consider a third dose if it will enable them to travel. A positive relationship has been shown between the intention to receive vaccines for travel reasons and vaccine safety and effectiveness perceptions, travel intention, and taking significant precautions when traveling [61]. Hence, we formulate the hypothesis as below.

**H1.** *Delegates’ trust in COVID-19 vaccines positively affects their involvement in venues*.

Previous studies have explored how the COVID-19 pandemic has changed customer consumption attitudes and decision behavior based on their economic orientation [57].

**H2.** *Delegates’ risk attitudes negatively affect their involvement in venues*.

#### 2.2.4. Satisfaction

Satisfaction is people’s feeling of pleasure or disappointment emerging after comparing the results of products with the expected outcome. If the outcome is below expectation, then customers’ satisfaction is considerably low. If the outcome meets their expectations, then their satisfaction is considerably high. Customers are highly satisfied and happy if outcomes exceed their expectations [62]. 

When performance differs from expectations, dissatisfaction results, and satisfaction represents a perceived bias, which is the difference between expectation and performance after consumption [63]. Satisfaction can also be considered as the level of people’s positive feelings about experiences [64]. In the tourism and hospitality sector, satisfaction first represents a function containing before- and after-travel experiences. Tourists are satisfied when their travelling experience outcomes are better than their expectations. However, they are dissatisfied when their travelling experience outcomes are not as good as their expectations [65]. The ordering of quality satisfaction is supported for events and festivals [66], and it has been confirmed in such fields as sports [67,68].

Previous studies have found that value and service quality perception impact satisfaction, and satisfaction impacts post-behaviors and loyalty [63,67,69,70,71,72,73,74,75]. The preceding studies have shown the contribution of satisfaction in their models. That is, attendees’ satisfaction, is a strong factor that impact attendees’ loyalty.

Research has reported that life satisfaction increased migration in OECD countries [76]. Individuals’ trust has also been reported to be an important factor because of increasing satisfaction of life in countries where they migrated [77]. Thus, the following hypothesis is formulated.

**H3.** *Delegates’ trust in COVID-19 vaccines positively affects their satisfaction in venues*.

Customer satisfaction and loyalty was explained by three theoretical foundations: customer involvement, satisfaction, and customer loyalty [78,79,80]. They suggested that involvement positively influences event quality perceptions and satisfaction [81]. Hence, the hypothesis is developed.

**H4.** *Delegates’ involvement positively affects their satisfaction with venues*.

Satisfaction is affected by attitude [82].

**H5.** *Delegates’ risk attitude negatively affects their satisfaction in venues*.

#### 2.2.5. Loyalty

Loyalty represents attendees’ commitment in the convention industry and loyalty has already been expressed previously [83]. Loyal attendees create positive word-of-mouth (WOM) recommendations and revisit conventions in the future. Loyalty is also a frequently studied factor in event and tourism research that has explored the intention to take part in conferences in future, and the rate of prior attendance as two loyalty indicators [84,85,86].

Loyalty is an important factor based on measures of individuals’ attitude and behavior. A measure of individuals’ attitudes shows a unique relationship with service providers, exhibiting the idea of repeated patronage. In actual situations, actions involving individuals’ loyalty is difficult to determine, and many studies have used behavioural intentions, such as conative loyalty, to measure loyalty of action [87]. Tourists’ intentions to travel to destinations again and recommendation willingness often reflect the degree of destination loyalty [71,88].

Levels of satisfaction, evaluation of performance, and re-travel intention motivate normal involvement in the convention industry. Severt found numerous factors in examining WOM, such as satisfaction and re-travel intention following the first industry convention [89]. Severt found that attendees’ high satisfaction with experiences positively impacts their recommendation of and re-engagement with the conference [89]. Tanford explored attendees’ perspectives that impacted their loyalty to a large international convention and found that switching cost is the most significant factor that impacts the emotional commitment of attendees [90]. Locke explored the MICE literature in New Zealand using a content analysis method to develop a situational analysis framework that includes recommendation intentions, levels of satisfaction, and revisit intentions [86,91].

Trust engages the relationship between customers and service providers, particularly in terms of service quality [92]. Perceptions of government services and individual organizations will engage the attraction of tourists, including their understanding of destinations [93]. Hence, commitment, trust, and satisfaction influence tourist loyalty [94,95]. We formulate the following hypothesis based on the previous discussions.

**H6.** *Delegates’ trust in COVID-19 vaccines positively affects their loyalty in venues*.

Convention management studies have frequently reported the relationship between satisfaction and loyalty. From the marketing angle, attendee satisfaction is among the crucial strategies, and focusing on attendees’ requirements efficiently is important to convention industry success [89]. In most studies, attendees’ revisit intention decides their loyalty in the conventions sector [89,96], which is likewise considered attitudinal loyalty [97], behavioural intention [98], future intention [99], and willingness of stay [100]. Convention literature reviews have supported the relationship between satisfaction and loyalty. Hospitality management research has extensive documentation of the relationship between the two factors [101]. Therefore, the hypothesis is formulated [102].

**H7.** *Delegates’ satisfaction positively affects their loyalty to venues*.

Given that people have realized that going to public areas will increase the probability of COVID-19 infection, the pandemic has shown that the corresponding risk attitude is a significant factor that impacts clients [103]. Therefore, we formulate the following hypotheses.

**H8.** *Delegates’ risk attitude negatively affects their loyalty to venues*.

#### 2.2.6. Moderator of Trust

Koller argued that trust is a function of the degree of risk inherent in a situation [104]. Trust is significantly important with the presence of two situational factors: incomplete information asymmetry of product and risk [105]. Trust is the desire and willingness to rely on other parties in risk [106]. Prior studies have shown that the relationship between the definite behavior of consumers and intention is reliable, but using some moderators can support this relationship [107,108,109]. In the MICE industry, only limited research has measured how trust in COVID-19 vaccination impacts attendees’ attitude and behavior. Hence, the following hypotheses are developed.

**H9a.** *Delegates’ trust in COVID-19 vaccines moderates risk attitude and involvement*.

**H9b.** *Delegates’ trust in COVID-19 vaccines moderates risk attitude and satisfaction*.

**H9c.** *Delegates’ trust in COVID-19 vaccines moderates risk attitude and loyalty*.

Based on all the hypotheses above, the model is constructed as Figure 1:

## 3. Methodology

### Survey Design and Data Collection

In this study, we used a 5-point Likert scale (1 = strongly disagree, 5 = strongly agree) to construct a questionnaire based on existing research. Structural equation estimation and data were measured using SPSS (IBM Corp., Armonk, NY, USA) and Smart PLS (IBM Corp., Armonk, NY, USA). The scale is described as follows (See Appendix A):

Trust in vaccines is measured by three items [110]. A sample item is “I would take a COVID-19 vaccine if it were offered to me”.

Risk attitude is measured by three items [111]. The item is “I will not consider have meals with relatives and local friends after they took part in Macau MIF”.

Involvement is measured by six items adopted from Oppermann [112]. A sample item is “I feel Macau MIF is concern to me”.

Satisfaction is measured by five items adopted from Michelini and M. del C. de Rojas and Camarero [67,113]. A sample item is as follows: “MIF is one of the best places that I could have visited”.

Loyalty is measured by four items [86]. A sample item is “I would recommend the Macau MIF to my friends”.

All questionnaire items were back-translated to Chinese. The original questionnaire was part of a pilot test prior to the revision of the actual survey. In the pilot test, 136 individuals (most were students) were invited to participate online. The targets are typical MIF attendees in Macau, and 250 surveys are distributed in filed occupations of MIF, and 263 are from the online survey. Under the normalization of COVID-19, the field investigation that was applied for MIF only approving two days on 11 and 12 December. The original surveys were about 390, and 140 contained many mistakes in the filling out. The faulty surveys were removed. The online surveys were 300 which were distributed by friends and relatives who participated the MIF at Macau. A further 37 faulty surveys were removed. 

## 4. Data Analysis

### 4.1. Demographic Information

Table 1 shows the demographic information of the convention delegates. More female delegates (57.12%) than male delegates (42.88%) completed the survey. Age ranges, namely, 18–25 (34.11%) and 26–35 (32.94%), were nearly equal. In the past year, 66.08% of the delegates came to Macau less than three times. The main local delegates and those from mainland China comprised 50.88% and 48.34%, respectively.

### 4.2. Measurement Model

This research performed confirmatory factor analysis (CFA) using SPSS, Amos, and Smart PLS. CFA showed convergent, reliability, and discriminant validities. Cronbach’s alpha values were between 0.846 and 0.929, as shown in Table 2. This outcome approaches the minimum requirement according to [63]. The validity of convergence is acceptable. Loadings of factors are dissimilar from zero in significance, and the range is from 0.646 to 0.986.

CFA (*n* = 513) is for the constructed model. Normed fit index (NFI), goodness-of-fit index (GFI), adjusted goodness-of-fit index (AGFI), root mean squared error approximation (RMSEA), minimum discrepancy per degree of freedom (CMIN/DF) and comparative fit index (CFI) are standard measurements to test the fitting in CFA. According to the general standard, values below 0.01, 0.05, and 0.08 are good, mediocre fit, and perfect, respectively, for RMSEA [114]. Moreover, the model indicates a poor situation if RMSEA > 0.1. The sample data and hypothetical model show acceptable fit if CMIN/DF < 3 [115], and reasonable fit if CMIN/DF < 5 [116].

When the standard values approach the good level of fit between the testing model and the data, AGFI, GFI, NFI, and CFI follow 0.8 < AGFI < 0.9, 0.85 < GFI, NFI < 0.9, CFI > 0.9, proving that the model is acceptable [117,118,119]. Table 3 shows the values of the empirical study: RMSEA = 0.06, AGFI = 0.886, NFI = 0.936, CFI = 0.958, and CMIN/DF = 2.827. That is, model fitting is generally acceptable. The Fornell–Larcker criterion shown in Table 4 indicates that the measurement model has been confirmed as reliable and valid.

Table 5 shows the routes of each dimension. Routes analysis shows that trust in vaccines has a significant moderating effect on the relationship between risk attitude and satisfaction. Thus, H9b is supported. Trust in vaccines has a significant positive effect on involvement. Therefore, H1 is supported.

Risk attitude negatively affects involvement, satisfaction, and loyalty significantly, which separately supports hypotheses H3, H4 and H5. Involvement positively affects satisfaction significantly, which supports hypothesis H7. Satisfaction positively affects loyalty significantly, which supports hypothesis H8.

## 5. Discussion

The main purpose of this article was to explore the moderating effect of trust in COVID-19 vaccination on delegates’ risk attitudes and behaviour in relation to conventions. 

The trust in COVID-19 vaccination moderated the relationship between risk attitude and satisfaction. However, trust in vaccination was not significant in moderating the relationship between delegates’ risk attitude and involvement in convention activities. Risk attitude was found to negatively impact delegates’ involvement, satisfaction, and loyalty in convention activities. When risk attitude increases, delegates’ involvement, satisfaction and loyalty decrease. The results support the findings of previous studies [57,82,103]. Involvement positively impacts delegates’ satisfaction. Satisfaction positively impacts delegates’ loyalty. When the level of involvement increases, the satisfaction increases, and the revisit intention increases. The results support the findings of previous studies [81,102].

The above result may be explained as below.

When people’s trust in vaccination is higher, it reduces the peoples’ risk attitude to COVID-19, and lower risk attitude also causes people to more easily experience higher satisfaction when participating in the activities. For improving participation, the organizers can arrange special convention activities for the delegates who have received COVID-19 vaccination. In actual situations, delegates’ participation would be considered more about the attitude to the COVID-19-related environment (local anti-COVID-19 policy, local government warning information) than COVID-19 vaccination propaganda. Whether or not delegates would like to participate in convention activities again depends on their attitude to the COVID-19 environment. Delegates considered more of the risk under COVID-19 environment, and they would be more careful in participating in activities. The same reason also caused delegates to have lower passion with regard to convention satisfaction and loyalty. Delegates have good experiences in participating in convention activities, thereby creating higher satisfaction with the activities and making them come back to the activities again.

### 5.1. Theoretical Contributions

The main contribution is the construction of a model based on trust in COVID-19 vaccination. The trust factor construct is complex and focuses on the relationship between organizations and individuals [36].

From the psychological perspective, the outcome shows how trust factor moderates individuals’ decisions to travel. This study also explored several mediators of trust in vaccination, thereby affecting other factors indirectly: risk attitudes, involvement, satisfaction, and loyalty. The outcomes indicate that peoples’ involvement is affected by trust in COVID-19 vaccination, and the other three variables are also affected by moderating trust in vaccination. Risk attitudes and trust in vaccinations are two important factors explored in this research. Exploring delegates’ behavior contributes to the tourism literature.

Lastly, Macau enjoys strategic advantages in developing the MICE industry [120]. However, most previous studies on the MICE industry have focused on self-development, convenience, personal attraction, activities and educational benefits [121], or even on technical support, cultivation and development exhibition, and developing markets [122]. Only a few studies have concentrated on trust in vaccination affecting convention development in Macau during the COVID-19 pandemic.

### 5.2. Practical Contributions

From a practical point of view, this study offers an empirical investigation of how trust in COVID-19 vaccination moderates delegates’ involvement, satisfaction, and loyalty via risk attitude. The findings of this study are that trust in vaccination moderates the connection between risk attitude and satisfaction, and trust in COVID-19 vaccination directly affects involvement.

Based on above, the government should offer detailed information on vaccination safety to enable people to understand; in particular, those belonging to the low-risk-tolerance group should be prioritized [123]. Social media, health information centres, and advertisements on public transportation can be channels to disseminate safety information on COVID-19 vaccination. Individuals’ perception of risk will be improved by enhancing public safety. The level of local safety will cause high exaggeration among local guides and operators. To reduce some unnecessary information on high-risk activities or attractions, they can advertise products by repackaging [124]. Meanwhile, delegates should actively obtain pandemic risk and vaccination information from other channels. Unbiased and professional operators can offer precise information to reduce misperception and increase security. The government ought to make a long-term plan for supporting the introduction or cultivation of MICE industry experts and talents [125]. 

## 6. Conclusions

Although conventions may still be limited owing to the COVID-19 pandemic, trust in vaccines appears to be an important factor to boost Macau tourism industry confidence. Unbiased and professional operators can offer precise information to reduce misperception and increase the security of the Macau MICE industry. This research focused on exploring the dimensions that could impact the involvement, satisfaction, and loyalty to the Macau convention industry. The COVID-19 pandemic has resulted in considerable fear in society. In this research, young delegates were interested in this investigation, including most of those aged 18–25 years (34.11%) and 26–35 years (32.94%). The satisfaction and loyalty relied considerably on their participation and COVID-19 risk attitude. They are active participants in convention activities, and not sensitive about COVID-19.

Trust theory, TPB, and standard satisfaction–loyalty theory provided bases for the research model used in this study. This paper hypothesized that trust in vaccination and risk attitude influence delegates’ involvement in convention venues, and also hypothesized that trust in vaccination moderates each relationship of two factors (risk attitude–involvement, risk attitude–satisfaction, risk attitude–loyalty). The outcomes show that delegates in Macau appeared to be more concerned with participation, which would influence their satisfaction of convention experience and whether or not they would come to the convention again. Trust in vaccination influences their behavior in the convention but would not influence their satisfaction and loyalty in convention activities. This research provides theoretical and practical contributions as follows.

In future study, we should investigate factors such as motivation, lifestyle, and personality. Different attitudes to participation in convention activities exist in the West and East, and performing the related comparative study would have been meaningful. Lastly, increasing the sample size should also be considered in the future for representativeness.

## Figures and Tables

**Figure 1 ijerph-20-03936-f001:**
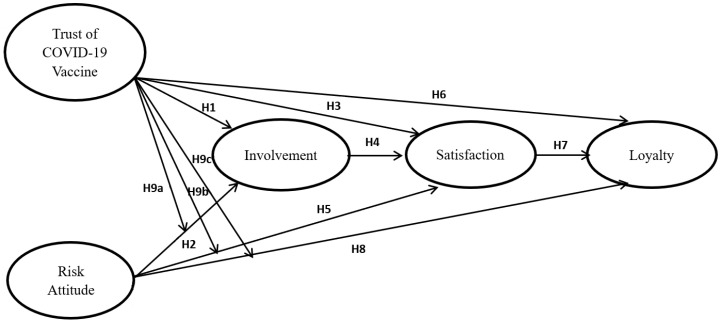
The Research Model.

**Table 1 ijerph-20-03936-t001:** Demographic Profile of Visitor.

Variable		Frequency (*n* = 513)	Percentage (%)
Gender	Male	220	42.88
	Female	293	57.12
Age	18–25	175	34.11
	26–35	169	32.94
	36–45	86	16.76
	46–55	49	9.55
	above 56	34	6.63
Visit times in past 1 year	less 3 times	339	66.08
	3–5 times	174	33.92
From	Macau	261	50.88
	Hongkong	4	0.78
	Mainland China	248	48.34

**Table 2 ijerph-20-03936-t002:** Reliability, CR, and AVE.

Variable	Item	Factor Loading	Cronbach’s Alpha	Component Reliability (CR)	Average Variance Extracted (AVE)
Trust of Vaccines	I would take a COVID19 vaccine if it were offered to me	0.651	0.86	0.866	0.689
	I believe it works	0.819			
	I believe healthcare provider about the risks and benefits with coronavirus vaccines	0.986			
Risk Attitude	Under the normal circumstances of the epidemic, I cannot accept participating in exhibition activities with my family	0.932	0.929	0.955	0.875
	Under the normal circumstances of the epidemic, I cannot accept participating in exhibition activities with local friends or relatives	0.950			
	After participating in exhibition activities, I cannot accept dining with local friends or relatives	0.925			
Involvement	I feel the MIF is concern to me	0.660	0.903	0.925	0.676
	The MIF activity is beneficial to me	0.802			
	The MIF activity is appealing to me	0.848			
	It is very significant to me	0.875			
	I am very interested in MIF activity	0.875			
	I feel exciting in MIF activity	0.852			
Satisfaction	MICE venues are one of the best options I can visit	0.804	0.846	0.891	0.625
	I am very happy to be able to participate in the exhibition activities	0.887			
	I had a good time and had a lot of fun at the exhibition	0.875			
	Under the normalization of the epidemic, many people still participate in exhibition activities	0.646			
	I am very reassured by the measures to control the epidemic during the exhibition activities	0.713			
Loyalty	I will recommend my friends to participate in exhibition activities	0.919	0.903	0.933	0.777
	I will encourage my friends to participate in exhibition activities	0.919			
	I will promote the positive meaning of the exhibition to others	0.867			
	Even if there is an epidemic, I am happy to participate in exhibition activities	0.816			

**Table 3 ijerph-20-03936-t003:** CFA standard values.

Model	RMSEA	CFI	AGFI	GFI	AGFI	CMIN/DF
Default model	0.06	0.958	0.886	0.913	0.886	2.827
Saturated model		1		1		
Independence model	0.267	0	0.153	0.23	0.153	37.508

**Table 4 ijerph-20-03936-t004:** Fornell-Larcker criterion.

	Trust of Vaccines	Risk Attitude	Involvement	Satisfaction	Loyalty
Trust in Vaccines	0.83				
Risk Attitude	0.181	0.935			
Involvement	0.067	−0.045	0.822		
Satisfaction	0.052	−1.56	0.733	0.791	
Loyalty	0.01	−0.264	0.641	0.783	0.881

**Table 5 ijerph-20-03936-t005:** Route Analysis.

	Initiative Sample (O)	Mean of Sample (M)	Standard Deviation (STDEV)	*t*-Test (O/STDEV)	*p* Value
**I** **nvolvement** **-> S** **atisfaction**	0.743	0.743	0.023	32.242	0.000
**Risk Attitude** **-> I** **nvolvement**	−0.100	−0.099	0.048	2.093	0.036
**Risk Attitude** **->** **Loyalty**	−0.155	−0.157	0.031	5.068	0.000
**Risk Attitude** **-> S** **atisfaction**	−0.106	−0.107	0.033	3.267	0.001
**S** **atisfaction** **->** **Loyalty**	0.725	0.725	0.028	26.169	0.000
**Trust of Vaccines -> I** **nvolvement**	0.152	0.154	0.065	2.344	0.019
**Trust Of Vaccines ->** **Loyalty**	−0.013	−0.012	0.032	0.391	0.696
**Trust of Vaccines -> Satisfaction**	0.013	0.017	0.035	0.382	0.702
**Moderator a-> I** **nvolvement**	−0.005	−0.002	0.049	0.103	0.918
**Moderator b -> S** **atisfaction**	0.070	0.067	0.033	2.104	0.035
**Moderator c ->** **Loyalty**	0.051	0.048	0.032	1.581	0.114

## Data Availability

This research data is the first hand data which is survied from The 26th Macao International Trade and Investment Fair.

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
