# Peer review of "Understand Delegates Risk Attitudes and Behaviour: The Moderating Effect of Trust in COVID-19 Vaccination"

_ijerph, 2023, doi:10.3390/ijerph20053936_

Round 1

Reviewer 1 Report

In the current manuscript, the authors have explored how delegates’ trust in vaccination may impact involvement, satisfaction, and loyalty in conventions.

The article is well structured into sections and subsections. It is within the scope of the journal. However, there are some major concerns that need to be addressed to improve the article. There are many sentences throughout the manuscript that are confusing and lack clarity. The detailed comments are below:

1)     The rationale of conducting the study is not clear. The role of trust in the likelihood of receiving a COVID-19 vaccine and subsequent impacts are already available in literature.

2)     The title does not reflect the content of the study.

3)     The sentences in the Abstract (Page 1, line 9) contradict the sentences in the Discussion (Page 9, line 357-359).

4)     Page 1, line 13-15: Based on line 9, if trust has no impact on convention loyalty and satisfaction, then, why there is a need to boost delegates confidence?

5)     Page 1, line 15: It is not clear what is recommended for the convention industry.

6)     Page 1, line 25-27: “…a drop of 36.55%”, compared to what (previous year)?

Similarly, “…decreased by 17.85%”, compared to what?

7)     Page 1, line 28-29: “Turnover loss of the MICE sector will decrease by at least 40% in 2020…”. It is already 2023, the author can present updated information till 2022.

8)     Page 1, line 34-37: The sentence requires rephrasing.

9)     Page 2, line 62: There is a hyperlink present, that needs to be removed.

10)  Page 9, Table 4: The presentation of values needs to be consistent. In some places zero is missing before the decimal point.

Moreover, ** symbol is present in some places, but it is not mentioned what is signifies.

11)  Page 9, lines 357-359 and lines 363-365: The sentences contradict each other. Rephrasing is required to bring clarity to the readers.

12)  Page 10, line 380-381: Here, authors mention that trust in vaccine appears to be an important factor to boost local economies and the tourism industry. This is contrary to what is conveyed in the abstract lines 9-12.

13)  Page 10, lines 385-387: Only young delegates being interested in the investigation does not necessarily mean that older delegates do not trust the effects of vaccination.

14)  Page 10, lines 415-416: The sentence needs rephrasing for clarity.

15)  Reference section: Some references have missing information or special characters which need to be corrected. Check reference 58, 60, 62, 63, 91, 92,103, and 114.

Author Response

Dear Reviewer

   How are you?

   The detailed information on the revision has bee uploaded.

   Thank you very much.

Cheers

Songhong Chen

Reviewer 2 Report

Dear authors,

I have several suggestions:

1. Reconsider the title- it is not clear enough.

2. Check the punctuation. I have noticed that there is no space between "." and the new sentence. Use space where it should be space. There are many examples for bad punctuation in the text.

Example: line 22 "MICE(such as: meetings, incentives, con..."

Example: line 25 "countries [5].According to..."

3. I have seen that you have written in the text "COVID-19", but also "covid19". Use one abbreviation.

4. Explain the meaning in the abbreviation "COVID-19", when you mention it for the firs time in the text.

5. I have noticed there is a hyperlink in the text- remove it. Line 63 " Jinge Yao (2021)" .

6. Section "Literature review"- it is not described well. How you have performed it? You can follow the prism guidelines for better understanding.

7. The fonts size of the text in" table 1", " table 2", "table 3", "table 4" and "table 5"  is different. Correct it.

8. The discussion is too short and not clear. Revise it.

Author Response

Dear Reviewer 

How are you?

The revision of this article has been uploaded.

Thank you very much.

Cheers

Songhong Chen

Reviewer 3 Report

In this manuscript, Chen, S et al., attempted to understand the role of trust in getting COVID-19 vaccines and its effect on people’s risk taking and social behaviors among those who attended the conventions in Macao.  A survey result with 514 participants was used for data analysis to test whether trust in vaccine had impact on risk taking, loyalty of convention attendance, and related feeling of satisfaction by attending the conference, etcc.. 

The topic is an interesting oneHowever, the manuscript is missing some key information for this reviewer to understand what was asked in the survey therefore to judge whether the conclusion could be supported, let alone that some places of writing need to be clarified. 

  1. The analysis of a survey result is dependent upon the survey questions designedPlease include the survey questions in the manuscript, not just example questionsAlso, since there are several COVID-19 vaccines available on the market, which ones does this survey is asking the “trust” aboutWill different vaccine generate different trust issues? 

  1. If the effect of trust with different vaccines can be normalized by your analytical method, what is the conclusion thenIf it cannot be normalized, what will the conclusion change to?  

  1. Here are some examples of places that the writing should be revised: 

Title: “risk rule” 

Abstract: what is “others significant”? 

Abstract: please clarify the last sentence 

Introduction: please define TPB when you first used this abbreviation 

Etc... 

Author Response

Dear Reviewer

How are you?

The revision of the manuscript report has been uploaded.

Thank you very much.

Cheers

Songhong Chen

Round 2

Reviewer 1 Report

The revised version of the manuscript has significantly improved. However, in some places, the sentence formation requires improvement, and some grammatical errors need correction.

1)     The sentences in the Abstract (Page 1, line 11): “Risk attitude negatively effects on involvement, satisfaction, and loyalty in significantly.” The sentences require rephrasing to improve clarity.

2)     Page 1, line 16-17: There is typing error, “…and to increase security level.”

3)     Page 1, line 24-25: There are grammatical errors in the sentences.

4)     Page 1, line 27-28: The sentence formation needs improvement for clarity. 4843 represents delegates?

5)     Page 1, line 38-39: The sentence formation needs improvement.

6)     Page 10, line 339-342: The sentences require rephrasing to improve clarity.

7)     Page 10, line 353: If involvement positively impacts delegates’ satisfaction. Then the following sentence would be, “When the level of involvement increases, the satisfaction increases…”.

8)     Page 10, line 359-261: The sentence needs improvement for clarity.

9)     Page 11, line 405, 412: There is typing error. “…and to increase security level.”

Author Response

Dear Reviewer

We are grateful for the various constructive suggestions. The thorough review process has indeed improved our study of which we are very appreciative. Responses are addressed in blue highlights words according to the reviewers’ comments and summarized as below.

  • The sentences in the Abstract (Page 1, line 11): “Risk attitude negatively effects on involvement, satisfaction, and loyalty in significantly.” The sentences require rephrasing to improve clarity.

Thank you for the reviewers’ comments. The expression has been revised in blue highlight of the article. Please find it as below.

  • Page 1, line 16-17: There is typing error, “…and to increase security level.”

Thank you for the reviewers’ comments. The sentence has been revised in blue highlight of the article. Please find it as below.

  • Page 1, line 24-25: There are grammatical errors in the sentences.

Thank you for the reviewers’ comments. The sentence has been revised in blue highlight of the article. Please find it as below.

  • Page 1, line 27-28: The sentence formation needs improvement for clarity. 4843 represents delegates?

Thank you for the reviewers’ comments. The sentence has been revised in blue highlight of the article.

  • Page 1, line 38-39: The sentence formation needs improvement.

Thank you for the reviewers’ comments. The sentence has been revised in blue highlight of the article.

  • Page 10, line 339-342: The sentences require rephrasing to improve clarity.

Thank you for the reviewers’ comments. The sentence has been revised in blue highlight of the article.

  • Page 10, line 353: If involvement positively impacts delegates’ satisfaction. Then the following sentence would be, “When the level of involvement increases, the satisfaction increases…”.

Thank you for the reviewers’ comments very much. There is typing error. And the sentence has been revised in blue highlight of the article.

  • Page 10, line 359-261: The sentence needs improvement for clarity.

Thank you for the reviewers’ comments very much. The sentence has been improved that is in blue highlight.

  • Page 11, line 405, 412: There is typing error. “…and to increase security level.”

Thank you for the reviewers’ comments very much. The sentence has been improved that is in blue highlight.

Reviewer 2 Report

The revised version of the manuscript is better than the first version. 

Author Response

Dear Reviewer

Thank you very much for your comments.

Reviewer 3 Report

The revision has done well.

Author Response

(The authors gave the same response as above.)
